# Gut Microbiota Ecological and Functional Modulation in Post-Stroke Recovery Patients: An Italian Study

**DOI:** 10.3390/microorganisms12010037

**Published:** 2023-12-25

**Authors:** Riccardo Marsiglia, Chiara Marangelo, Pamela Vernocchi, Matteo Scanu, Stefania Pane, Alessandra Russo, Eleonora Guanziroli, Federica Del Chierico, Massimiliano Valeriani, Franco Molteni, Lorenza Putignani

**Affiliations:** 1Immunology, Rheumatology and Infectious Diseases Research Area, Unit of Human Microbiome, Bambino Gesù Children’s Hospital, IRCCS, 00146 Rome, Italy; riccardo.marsiglia@opbg.net (R.M.); chiara.marangelo@opbg.net (C.M.); pamela.vernocchi@opbg.net (P.V.); matteo.scanu@opbg.net (M.S.); federica.delchierico@opbg.net (F.D.C.); 2Unit of Microbiomics, Bambino Gesù Children’s Hospital, IRCCS, 00146 Rome, Italy; stefania.pane@opbg.net (S.P.); alessandra.russo@opbg.net (A.R.); 3Villa Beretta Rehabilitation Center, Valduce Hospital Como, 23845 Costa Masnaga, Italy; eleonora.guanziroli@gmail.com (E.G.); franco56.molteni@gmail.com (F.M.); 4Developmental Neurology, Bambino Gesù Children Hospital, IRCCS, 00165 Rome, Italy; massimiliano.valeriani@opbg.net; 5Center for Sensory Motor Interaction, Aalborg University, 9220 Aalborg, Denmark; 6Unit of Microbiomics and Research Unit of Human Microbiome, Bambino Gesù Children’s Hospital, IRCCS, 00146 Rome, Italy

**Keywords:** ischemic stroke, gut–brain axis, gut microbiota ecology, SCFAs, tryptophan derivatives, fecal zonulin

## Abstract

Ischemic stroke (IS) can be caused by perturbations of the gut–brain axis. An imbalance in the gut microbiota (GM), or dysbiosis, may be linked to several IS risk factors and can influence the brain through the production of different metabolites, such as short-chain fatty acids (SCFAs), indole and derivatives. This study examines ecological changes in the GM and its metabolic activities after stroke. Fecal samples of 10 IS patients were compared to 21 healthy controls (CTRLs). GM ecological profiles were generated via 16S rRNA taxonomy as functional profiles using metabolomics analysis performed with a gas chromatograph coupled to a mass spectrometer (GC-MS). Additionally fecal zonulin, a marker of gut permeability, was measured using an enzyme-linked immuno assay (ELISA). Data were analyzed using univariate and multivariate statistical analyses and correlated with clinical features and biochemical variables using correlation and nonparametric tests. Metabolomic analyses, carried out on a subject subgroup, revealed a high concentration of fecal metabolites, such as SCFAs, in the GM of IS patients, which was corroborated by the enrichment of SCFA-producing bacterial genera such as *Bacteroides*, Christensellaceae, *Alistipes* and *Akkermansia*. Conversely, indole and 3-methyl indole (skatole) decreased compared to a subset of six CTRLs. This study illustrates how IS might affect the gut microbial milieu and may suggest potential microbial and metabolic biomarkers of IS. Expanded populations of *Akkermansia* and enrichment of acetic acid could be considered potential disease phenotype signatures.

## 1. Introduction

The gut–brain axis is a bidirectional communication system between the gut and the brain. In particular, it involves the central nervous system (CNS), the autonomous nervous system (ANS), the enteric nervous system (ENS) [1] and the gut microbiota (GM), the latter including all microorganisms (bacteria, fungi, archaea and viruses) that populate the human gastrointestinal (GI) tract [2]. The CNS and the GM have multiple avenues of communication, including (i) neural (the vagal nerve); (ii) immune (cytokines) and (iii) metabolic (short-chain fatty acids, SCFAs) pathways. Perturbation of this axis can be involved in neurodegenerative disorders [3]. In particular, “top-down” signaling, from CNS to the gut, can influence the ANS directly, via the sympathetic and parasympathetic systems, or indirectly, through the modulation of the enteric nervous system [4]. These nervous signals can influence motility, permeability, microbiota composition and other intestinal functions [5]. Conversely, the GM can influence the brain through “bottom-up” signaling. Neurotransmitters and signaling molecules, such as SCFAs, γ-amino-butyric acid, tryptophan derivatives, catecholamines and bile acid metabolites, can interact both with receptors in the gut wall and the ENS to communicate to the brain through vagal afferents [6,7]. The GM is closely related to human health and is composed of a multitude of different bacterial species, mostly belonging to six main phyla, i.e., Firmicutes, Bacteroidetes, Actinobacteria, Proteobacteria, Fusobacteria and Verrucomicrobia [8]. Under normal circumstances, the GM performs many crucial physiological functions, such as the maintenance of gut integrity [9], the production of SCFAs via fermentation of complex polysaccharides, the biosynthesis of essential amino acids and vitamins [10], and protection against pathogens by regulating host immunity [11]. On the other hand, an imbalance within the GM (i.e., dysbiosis) is related to numerous GI and metabolic diseases, including diabetes, dyslipidemia and obesity [12], which along with hypertension and atherosclerosis represent important risk factors for ischemic stroke (IS).

IS occurs when an obstruction leads to substantial decrease in blood flow to the brain [13]. Furthermore, IS can lead to alterations in the composition of the GM and its functions, and, conversely, GM alterations and changes in GM-derived metabolites have been associated with IS risk [14]. Several studies have linked patient prognosis following IS with GM dysbiosis [15] and altered SCFA production, which in turn have numerous homeostatic and anti-inflammatory effects related to GM biochemistry [16].

IS can occur at any age or stage of life. While the incidence of pediatric stroke is considerably lower than it is in adults, the neurological consequences [17] showing an impact on life quality and on the health care service is significantly higher following pediatric stroke [18].

The aim of this study was to examine changes in GM taxonomy and its activity in patients after a stroke event in order to identify biomarkers of disease and patient prognosis. Furthermore, this study aims to generate a disease-related GM model that could be transferred to pediatric populations.

## 2. Materials and Methods

### 2.1. Study Participants and Clinical Data

Ten patients with ischemic stroke (IS) aged 54–81 years (median, 70.5 years ± IQR 14) of whom there were 6 males and 4 females, were recruited at the Villa Beretta Rehabilitation Center, Valduce Hospital, Costa Masnaga, Lecco, Italy, between January and December 2021. Twenty-one age-matched healthy subjects (median, 66.0 years ± IQR 9.0), belonging to an epidemiological survey executed at the Human Microbiome Unit of Bambino Gesù Children’s Hospital in Rome (BBMRI Human Micro-biome Biobank, OPBG) were considered to constitute a sample and digital biobank of reference controls (CTRLs).

The majority of patients presented risk factors for ischemic stroke, including hypertension, diabetes, dyslipidemia and obesity (Appendix A). They received routine pharmacotherapy for controlling comorbidities such as amlodipine and metformin. Exclusion criteria for the study were: hemorrhagic stroke or hemorrhagic infarction, probiotics and antibiotics administration at least one month prior to sample collection, and GI, oncological or neurodegenerative diseases. The participants’ metadata, including clinical assessment to evaluate lower limb function and blood biochemical parameters (BBP) (alanine aminotransferase (ALT), aspartate aminotransferase (AST), gamma-glutamyl transpeptidase (GGT), triglycerides, cholesterol, glucose, albumin and bilirubin), were evaluated at T_0_ (beginning of rehabilitation period) and at T_1_ (6 weeks later) at the Rehabilitation Center. Anamnestic variables (age, BMI) were recorded at T_0_. After the stroke event, a clinical evaluation based on the International Classification of Functioning, Disability and Health (ICF) [19] was performed at T_0_ and T_1_ time points on both the affected and nonaffected sides, with different assays such as: the Box and Block test (BBT) to determine the number of blocks moved from one side to the other by the arm [20]; the Action Research Arm test (ARAT) to evaluate arm and hand functions [21]; the Motricity Index test (MI_TOT) to estimate lower extremities’ strength [22] and the 10-Meter Walking test (10 MWT) to evaluate walking speed over a 10 m distance [23].

In the current study, considering the small sample size, only the continuous variable MI_TOT was evaluated to assay the motricity on the affected side.

### 2.2. Ethics Statement

The study was approved by the Villa Beretta Ethics Committee for patients with ischemic stroke (protocol v.1.2 10/11/2020) and by the OPBG Ethics Committee for healthy subjects’ (1113_OPBG_2016) cohorts. Written informed consent was obtained from all participants.

### 2.3. Fecal Sample Collection

Ten fecal samples from IS patients were collected at T_0_ and T_1_ and immediately frozen at −20 °C at the Villa Beretta Rehabilitation Center, Valduce Hospital, and then delivered to the Human Microbiome Unit of Bambino Gesù Children’s Hospital in Rome, where they were stored at −80 °C until metataxonomy and metabolomics analyses and gut permeability assay were performed. Fecal samples from the age-matched CTRLs were available as aliquots at −80 °C at the OPBG BBMRI Biobank. Further ecological profiles were provided for the 10 IS patients and 21 CTRLs, the latter including 15/21 digital profiles already available as biobank content and 6/21 newly produced.

### 2.4. 16S rRNA Targeted Metataxonomy Analysis

The general workflow included DNA extraction using the QIAmp Fast DNA Stool mini kit from Qiagen, Hilden, Germany and amplification and sequencing of the variable region V3–V4 from the 16S rRNA gene, according to 16S Metagenomic Sequencing Library preparation protocol (Illumina, San Diego, CA, USA). The raw reads were analyzed by Quantitative Insights into Microbial Ecology software (QIIME 2 v2023.2) [24]. The QIIME2 plugin for DADA2 was used for quality control, denoising, chimera removal, trimming and construction of an amplicon sequence Variant (ASV) table.

The taxonomic analysis was performed using a naïve Bayes model pretrained on database Greengenes (v13.8, August 2013, https://greengenes.secondgenome.com/) through the QIIME2 plugin q2-features classifiers. Sequencing data associated with this study were uploaded to the NCBI bioproject database: PRJNA1030992 (IS patients), PRJNA996768 and PRJNA531579 (CTRLs).

### 2.5. Gas Chromatography Coupled to Mass Spectrometry (GC-MS) Metabolomics Analysis

Overall, 10 patient samples and the subset of available CTRL biobank samples (i.e., 6/21), were processed for indole, 3-methyl-indole (skatole) and SCFAs determinations.

Hence, SCFAs, indole and skatole concentrations were estimated by Gas Chromatograph (GC) 7890 A (Agilent Technologies, Santa Clara, CA, USA) coupled to a mass spectrometer (MS) 5977 (Agilent Technologies, Santa Clara, CA, USA) in electron impact mode (ionization voltage of 70 eV) equipped with a DB-HeavyWAX (60 m × 0.250 mm × 0.25 µm) capillary column Agilent Technologies. In particular:An aqueous stock standard solution was prepared from chemical reagents of acetic acid (99%), propanoic acid (99%) and butanoic acid (99%) purchased by Merck (Darmstadt, Germany) with a concentration of 200 mm for each acid. All the stock standard solutions were stored at −20 °C until used.For SCFAs analysis, fecal samples (250 mg) were added to 1.25 mL of H_2_O and homogenized in a shaker for 3 min. The samples were then acidified with HCl (3 M) to a pH of 2–3. Finally, they were centrifuged at 5000× *g* at 4 °C for 10 min. After centrifugation 1 mL of supernatant was centrifuged at 15,000× *g* at 4 °C for 15 min. The supernatant was transferred and basified at pH 6 with NaOH (5 M), after which hexanoic acid (final concentration of 0.4 mg/mL) was added as an internal standard before sample screening. The samples were then filtered through 0.45 µm and 0.22 µm filters. Finally, 1 µL of each sample was injected into GC-MS. The temperature program was: 50 °C for 2 min, a temperature increase of 10 °C per min at 70 °C, 3 °C per min at 85 °C, 5 °C per min at 110°C, 15°C per min at 200 °C, 20 °C per min at 230 °C, 10 °C per min at 240 °C, then 240 °C for 5 min, according to the adapted *in house* procedures performed by Zhao et al. [25]. The method was validated by *in house* procedures according to European Medicines Agency (EMEA) and FDA guidelines [26,27] on bioanalytical method validation. The linearity of the calibration curve for each SCFA was assessed by the calculation of the coefficient of determination (R^2^). Validation was based on coefficient of variability (CV) of inter- and intraday reproducibility (%), limit of detection (LOD), limit of quantification (LOQ) and % of recovery.For indole and skatole determinations, fecal samples (100–500 mg) were managed and analyzed as previously described by Vernocchi et al. [28] using GC-MS coupled to solid-phase microextraction (SPME).

Compound identification was confirmed by injection of pure standards and a comparison of the retention time and corresponding MS spectra. Moreover, corroboration of the identification of all molecules was conducted by searching mass spectra in the National Institute of Standards and Technology (NIST, Rockville, MD, version 2d, build 26 April 2005) library and in the literature. Chromatograms were integrated and the quantitative data, expressed in mg/kg, were obtained by interpolation of the relative areas and the area of the internal standard.

### 2.6. Fecal Markers of Intestinal Inflammation and Permeability (Zonulin)

Zonulin was detected using an ELISA kit (IDK Zonulin ELISA for the in vitro determination of zonulin family peptides, ZFP, in stool, Immunodiagnostik AG, Bensheim, Germany) according to the manufacturer’s protocol using 15 mg of patient stool sample. Briefly, a biotinylated zonulin tracer was added to samples, standards, and controls. The treated samples, standards, and controls were incubated in microtiter plates coated with anti-zonulin polyclonal antibodies for one hour. Samples were then washed five times in 250 µL wash buffer, and the remaining liquid was absorbed. The biotinylated zonulin tracer is associated with peroxidase and can be found in streptavidin (100 µL), which was subsequently added to each well. The microplate was incubated, washed, and drained as described previously. The samples were then incubated with 100 µL substrate solution for 15 min, after which the reaction interrupted with an acidic stop solution. The absorption was determined immediately at 450 nm against a reference wave of 620 nm using an Infinite^®^ F50 compact ELISA absorbance microplate reader (TECAN, Männedorf, Switzerland). The quantitative data were represented by zonulin concentrations in ng/mL.

### 2.7. Statistical Analyses Processing

Firstly, longitudinal IS GM ecology, compared to CTRLs, was investigated at T_0_ and T_1_. Removal of unassigned ASVs and rarefaction of samples was performed. ASV table was filtered out, retaining the ASVs present in at least the 25% of the total samples. Moreover, a filter based on ASVs with relative abundance >1% was applied. A filtered matrix was selected for applying statistical analysis. The GM composition of patient and CTRL groups were analyzed and α and β diversity were performed by R “Phyloseq” package. The α diversity was calculated with R, using Shannon, Simpson and Chao1 indexes and Kruskal–Wallis and Mann–Whitney tests were applied for group comparisons. Principal coordinate analyses (PCoA) plots were constructed to illustrate the β-diversity algorithm based on the Bray–Curtis index. To test the association between the covariates and β-diversity algorithms, permutational analysis of variance (PERMANOVA) was used [29]. Targeted-metagenomics data were analyzed using supervised partial least-squares-discriminant analysis (PLS-DA) performed with the R “mixOmics” package for Bioconductor [30]. For metataxonomy (ASVs), metabolomic (SCFAs) and fecal marker (zonulin) results, the data normality distribution was evaluated using the Shapiro–Wilk test and non-parametric (Mann–Whitney U-test and Kruskal–Wallis, comparing population medians) and parametric test (*t*-test comparing population means) were used. Statistical significance was assessed at *p*-value ≤ 0.05 corrected by Benjamini–Hochberg FDR procedure [31]. The relative abundance of taxa in IS patients and CTRLs was represented with bar plot (R “ggplot2” package). The functional analysis was performed using Phylogenetic Investigation of Communities by Reconstruction of Unobserved States (PICRUSt2) [32]. Linear discriminant analysis effect size (LEfSe) was used to reveal the direction of pathways statistically significant between IS patients and CTRLs. To obtain KEGG description, the online database https://www.genome.jp/kegg/ko.html was used [33].

To correlate microbial taxa and SCFAs, Spearman correlation coefficients were exploited and corresponding *p*-values (corr.test function in the R “psych” package) were represented by a heatmap (R “pheatmap” package) [34].

Moreover, to evaluate associations between pairs of variables (i.e., metabolites, ASVs abundances, BBP and clinical variables), linear regression models and multivariable Pearson correlation tests were performed for continuous variables such as MI_TOT and BBP.

## 3. Results

### 3.1. Subject Characteristics, Anamnestic and Clinical Features

In this study, the 10 recruited IS patients were clinically evaluated. No significant differences between IS patient and CTRL sample groups were observed in age distribution (*p*-value = 0.095, non-parametric Mann–Whitney test). The majority of the IS patients presented risk factors for IS including hypertension (*n* = 8/10), diabetes (*n* = 2/10) and dyslipidemia (*n* = 6/10). Furthermore, at T_0,_ obese (*n* = 2/10) and overweight (*n* = 5/10) IS patients showed higher values of triglycerides, glucose and cholesterol compared to those with normal weight (*n* = 3/10) (Appendix A).

Neurological features regarding the evaluation of functional movement in IS patients (MI_TOT; BB; ARAT; 10 MWT) were reported in Appendix A. Of note, most IS patients (*n* = 7/10) at T_0_ were not able to walk and thus had a score of 10 MTW, whereas all patients started walking at T_1_.

Considering the BBT on the affected side, 5 out of 10 IS patients were able to move more than 23 blocks at T_0_, which increased to 8/10 at T_1_. Regarding ARAT of the affected side, 8 out of 10 IS patients showed score of <57 at T_0_, while the other 2 did not show any deficit. At T_1_, 3 out of 10 IS patients showed score of <57, while the other 7 showed the maximum score value of 57 (Appendix A).

The MI_TOT, evaluated on the only affected side, showed an increase at T1 (mean 69.8 ± SD 29.9) compared to T_0_ (mean 46.5 ± SD 29.4).

### 3.2. Gut Microbiota Metaxonomy

A total of 4428 sequences of 16S rRNA gene amplicons were obtained with a frequency average of 41,641 amplicons/sample and an average length of 406 nt (nucleotide) (calculated after primer removal). The GM composition of IS patients at T_0_ and T_1_ and CTRLs was analyzed by ecological analysis.

Firstly, the α diversity was performed in order to evaluate the GM diversity between CTRLs and IS patient groups. The α diversity, performed via Chao1 index, showed a significant difference between CTRLs and IS patients; in particular, CTRLs GM showed higher values of diversity, when compared to IS_GM, either at T_0_ and T_1_ (*p*-value = 0.042 and *p*-value = 0.038, respectively; Figure 1A). On the contrary, no statistically significant differences between patients and CTRLs, evaluated using Shannon and Simpson indexes, was observed.

Moreover, β diversity, performed by Bray–Curtis distance, showed IS patients (T_0_ and T_1_), grouped into two overlapping clusters, separated from CTRLs (*p*-value = 0.001, PERMANOVA; Figure 1B).

Also, the PLS-DA score plot (Figure 2A) showed clear group separation between CTRLs and IS patients. The loadings plot (Figure 2B) revealed that Coriobacteriaceae, Mogibacteriaceae, *Dorea* and Clostridiales discriminated the CTRLs, while *Phascolarctobacterium*, Christensenellaceae, *Sutterella* and *Alistipes* were associated with the IS patient group.

Consistent with previous findings, the GMs of both patients and CTRLs were composed of five main phyla: Firmicutes, Bacteroidetes, Proteobacteria, Actinobacteria and Verrucomicrobia. Particularly, Firmicutes and Bacteroidetes were the predominant phyla of both IS patients and CTRLs. However, the Verrucomicrobia and Proteobacteria phyla were significantly less abundant in CTRLs, while Actinobacteria phylum was more abundant in CTRLs than IS patients. Specifically, IS patients at T_1_ showed a significant increment (*p*-value FDR ≤ 0.01) of Verrucomicrobia compared to patients at T_0_ (Figure 3A).

The analysis of ASVs distribution at family level (L5) showed a significant higher abundance of Clostridiaceae (*p*-value FDR ≤ 0.05), Ruminococcaceae (*p*-value FDR ≤ 0.05) and Coriobacteriaceae (*p*-value FDR ≤ 0.01) in CTRLs compared to IS patients at both time points, whereas Verrucomicrobiaceae (*p*-value FDR ≤ 0.01) and Bacteroidaceae (*p*-value FDR ≤ 0.01) were significantly increased in IS patients cohort, specifically at T_1_ (Figure 3B). Conversely, Streptococcaceae (*p*-value FDR ≤ 0.01), Rikenellaceae (*p*-value FDR ≤ 0.01) and Enterobacteriaceae (*p*-value FDR ≤ 0.01) were significantly reduced in CTRLs compared to IS patients (Figure 3B).

At the genus level (L6), *Anaerostipes* (*p*-value FDR ≤ 0.05) and Clostridiales (*p*-value FDR ≤ 0.05) were significantly increased in the CTRLs (Figure 3C), while *Phascolarctobacterium (p*-value FDR ≤ 0.01), *Alistipes (p*-value FDR ≤ 0.001), Christensenellaceae (*p*-value FDR ≤ 0.01) and Sutterella (*p*-value FDR ≤ 0.001) were enriched in the GM of IS patients (Figure 3C). Moreover, *Akkermansia* (*p*-value FDR ≤ 0.05) and *Bacteroides* (*p*-value FDR ≤ 0.01) showed a significant decrement in the CTRLs compared to IS patients at both time points and these genera were more abundant in IS patients at T_1_ compared to IS_T_0_ (Appendix A). Furthermore, there were not statistically significant ASVs differences comparing IS patients at T_1_ and_T_0_ time point (Appendix A).

Focusing on the IS patients’ and CTRLs’ GM ecology comparison at T_0_, the LEfSE analysis corroborated the ASVs associated to the IS disease through the longitudinal time points, namely *Bacteroides*, *Akkermansia*, *Alistipes*, Christensellaceae (Appendix A). Moreover, the PICRUSt, predicted the main IS- and CTRL-related functional pathways. The LEfSe analysis showed 50 pathways differentially expressed (*p*-value ≤ 0.05). In particular, 33/50 were associated to IS, including lipopolysaccharide (LPS) biosynthesis and butanoate metabolism, while 17/50 to CTRLs (Appendix A).

### 3.3. Gut Microbiota Metabolome Characterization

The GM metabolome composition (SCFAs, indole and skatole) was analyzed on the subset of 20 IS patients at T_0_ and T_1_ (median, 70.5 years ± IQR 14) and 6 CTRLs (i.e., 6/21, median, 44.5 years ± IQR 30.25) by GC-MS/SPME analysis (Appendix A). Particularly, acetic acid showed highest values (Figure 4A), followed by butanoic (Figure 4B) and propanoic acid (Figure 4C).

In addition, by considering the comparison between the IS patients and CTRLs, acetic acid and butanoic acid were significantly increased in the IS GM patients at T_0_ and T_1_ compared to the CTRLs, while propanoic acid only showed significant differences at T_1_ compared to the CTRLs.

Moreover, skatole concentrations were under the Limit of Quantification (LOQ) for 5/10 IS_T_0_ and 6/10 IS_T_1_ patients, while the others showed lower levels compared to CTRLs (Appendix A).

Indole was only detected in the CTRL group (mean 287.2 mg/kg), while its concentration was under the LOQ in the IS patients (Appendix A).

### 3.4. Fecal Markers of Intestinal Permeability (Zonulin)

Regarding zonulin concentrations, higher values were observed for the IS patients at T_1_ compared to the IS patients at T_0_ (*p*-value ≤ 0.05) (Appendix A).

### 3.5. Correlation Analysis of Omics Data, Clinical Features and BBP

The Spearman correlation between SCFAs, zonulin and ASVs, showed in the heatmap (Figure 5), highlighted statistically significant (*p*-value ≤ 0.05) positive correlation, particularly between *Blautia* with acetic and butanoic acid; *Ruminococcus* (Ruminococcaceae) with acetic acid; *Streptococcus* with acetic, propanoic and butanoic acid; *Lachnospira* with butanoic and propanoic acid; *Catenibacterium* with butanoic acid. Furthermore, there were statistically significant (*p*-value ≤ 0.05) negative correlations, in particular between *Methanobrevibacter* with acetic and propanoic acid; *Desulfovibrio, Parabacteroides* and *Succinivibrio* with zonulin (Figure 5).

Moreover, the correlation between motricity index (MI_TOT), BBP (alanine aminotransferase (ALT), aspartate aminotransferase (AST), gamma-glutamyl transpeptidase (GGT), triglycerides, cholesterol, glucose, albumin and bilirubin) with the ASV, SCFAs and zonulin concentrations of the 10 IS patients at different time points were considered.

The correlation analysis between MI_TOT at time T_0_ and ASVs, SCFAs showed significant negative correlations with Christensenellaceae (R = −0.66, *p*-value = 0.038) and with acetic acid (R = −0.76, *p*-value = 0.01) (Appendix A). At time T_1_ MI_TOT was negatively correlated with *Bacteroides* (R = −0.73, *p*-value = 0.018), *Clostridium* (Lachnospiraceae) (R = −0.71, *p*-value = 0.022) (Appendix A), and no significant (*p*-value > 0.05) correlations with any SCFA.

In addition, the correlation analysis between BBP at T_0_ and T_1_ and ASVs (at L6) was performed (Appendix A). Particularly, by considering IS patients at T_0_ high positive and significant (*p*-value ≤ 0.05) correlations were observed: tryglicerides with *Akkermansia* and *Streptococcus*; AST with *Succinivibrio* and *Prevotella*; GGT with *Coprococcus*, *Alistipes*, *Butyricimonas* and *Sutterella*; glucose and albumin with *Streptococcus.* Conversely, negative and significant (*p*-value ≤ 0.05) correlations were observed as follows: ALT with Christensellaceae; tryglicerides with *Faecalibacterium;* glucose with *Megasphera* (Appendix A).

Considering clinical data of IS at T_1,_ high positive and significant (*p*-value ≤ 0.05) correlations were observed between AST and *Akkermansia*; cholesterol and *Ruminococcus* (Ruminococcaceae); glucose and *Alistipes*; bilirubin with *Anaerostipes*, Erysipelotrichaceae, *Eubacterium*, *Prevotella* and Enterobacteriaceae. On the other hand, ALT was negatively correlated with *Ruminococcus* (Ruminococcaceae) and *Prevotella* and there was negative significant (*p*-value ≤ 0.05) correlation between tryglicerides and Christensenellaceae (Appendix A). By performing the correlation analysis between BBP, SCFAs and zonulin related to IS patients at T_0_, positive correlations (*p*-value ≤ 0.05) were observed between zonulin with triglycerides and albumin (Table 1).

The BBP values in IS patients at T_1_ did not significantly correlate with SCFAs and zonulin (Appendix A).

## 4. Discussion

The aim of this study was to investigate the bidirectional communication along the gut–brain axis in IS patients. Dysfunction of the gut–brain axis, induced by dysbiosis, and GM variations are related to stroke risk factors such as obesity and atherosclerosis [35,36] and seem to be involved in the formation of atherosclerotic plaques [37]. Furthermore, some evidence suggests that stroke itself noticeably affects changes in the gut microbiota, and that these alterations, that could be linked to an increase in inflammation in different neurological diseases, in turn could influence stroke outcome [38].

Principally, it has been possible to observe GM taxonomy changes in IS patients after stroke compared to CTRLs, in particular with a decrease in Ruminococcaceae and Clostridiales (Clostridiaceae) and an increase in *Akkermansia*, *Bacteroides* and *Alistipes,* Christensenellaceae and *Sutterella*, recognized to be SCFA-producing bacteria, corroborated also by previous studies [39].

These microbes can act as pro- or anti-inflammatory agents, depending on different conditions. Several studies observed that these bacteria could ameliorate the symptoms of neuropsychiatric disorders by restoring the GM, mucosal barrier integrity and by modulating neuroinflammation [40,41,42]. Conversely, Bonnechère et al. showed that the genus *Akkermansia* was consistently enriched in the GMs of patients affected by neurological disorders, such as Parkinson’s disease, multiple sclerosis and stroke [38]. Moreover, *Akkermansia* has been also associated with diabetes and cardiovascular diseases, both common comorbidities of neurological disorders [38].

Furthermore, consistently with our results, Chen et al. [43] reported ischemic stroke triggering gut microbial alteration through enriching pathogens and opportunistic microorganisms such as SCFAs producers *Bacteroides* and *Alistipes* [44].

However, also an increment of *Bacteroides* and *Alistipes* has been observed in Alzheimer’s and Parkinson’s diseases, suggesting these genera as microbial potential indicators of CNS disorders [38,45].

In addition, previous studies have found an enrichment of Christensenellaceae, another SCFAs-producing genus, [46,47] in the gut of ischemic stroke patients [48].

Furthermore, *Sutterella*, an acetate producer, could be involved in neurodevelopmental disorders such as autism spectrum disorder (ASD) [49,50], gastrointestinal (GI) diseases, including inflammatory bowel disease (IBD) and coeliac disease (CD), and their comorbidities, such as cardiovascular disease and obesity [51,52].

After stroke, the noradrenaline release and the disruption of the axis between the gut and the vagus nerve may trigger GM modifications in injured patients, typically with a decrease in Ruminococcaceae [53,54,55], may be involved in the regulation of the immune system [56], as the same trend also occurred in IS patients in our study. Many studies examining changes in the GM of stroke patients have found a decrease in Clostridiaceae compared to health controls, consistent with these results. This reduction may be associated with an elevated inflammatory response or with infections post stroke, possibly related to serious brain injury and poor stroke outcome [14].

Moreover, following the ecological characterization of our cohort in which several SCFAs-producing bacteria were found, the prediction of functional profile showed several metabolic pathways associated with IS patients as butanoate metabolism and LPS biosynthesis pathways. Our results were in agreement with other evidence showing LPS synthesis significantly enriched in IS subjects [57]; also, Jia et al. [58] reported butanoate metabolism’s upregulation in high-risk IS subjects as observed in murine model.

Consistently with these findings, a higher SCFAs fecal concentration, specifically acetic, propanoic and butanoic acid was observed in our patients’ cohort after stroke event compared to CTRLs. SCFAs, as is well known, may have both beneficial or detrimental effects, in relation to their concentration, in different neurological diseases, as they are involved in gut–brain axis communication [51]. After stroke, SCFAs could have a role in the immune system, in particular on microglial activation and on T-cell recruitment to the infarcted brain [59]. Indeed, SCFAs have been shown to readily cross the blood–brain barrier and could induce beneficial therapeutic effects during the recovery period [60]. On the other hand, when present at high concentrations, as in our study, SCFAs may behave as neurotoxins passing through the mucosa layer and cell membranes where they could exert toxic effects [61] and may lead to neutrophil accumulation and exacerbation of inflammatory processes [62].

Moreover, high concentrations of acetic acid, the most expressed of all the SCFAs in this patients cohort, may be related to GM alterations as a consequence of acute brain injury [63]. Hence, several studies identified a dysregulation of metabolic activity of the GM in patients after stroke [12], as reported by Xu et al. [64]. This study found that cerebral ischemia could rapidly induce GM disorders and stimulate excessive nitrate production, which in turn could affect the anaerobic fermentation of carbohydrate degradation, leading to an increase in acetate production [65].

Previous evidences have found a progressive reduction in fecal SCFAs in elderly people [66,67], though this study revealed an increase in butanoic acid in elderly patients without any significant gut microbial changes. This could probably be associated with the stroke event, and a consequent increase in pro-inflammatory commensals to the detriment of beneficial microbes [68], but the potential clinical consequences remain to be fully elucidated due to high degree of variability of the human GM [68].

GM alterations are also known to be associated to motor functions and general disability level [69] as a consequence of a stroke event. In this study, elevated levels of SCFAs were detected in IS patients with limited motricity, which could be correlated with a decrease in intestinal contractile activity caused by inflammation secondary to the injury [70].

Zonulin, an indirect intestinal inflammatory biomarker, could also be associated with stroke and chronic inflammation [71]. Moreover, other studies have shown increased levels of zonulin in older adults with frailty [72]. In this work, there were significant differences in fecal zonulin concentrations between IS patients at different time points, as also observed by Zheng et al. [73], which probably reflects an intestinal or systemic inflammatory condition. In addition, the role of tryptophan derivatives, such as indole and skatole, emerged as important intermediaries in the microbiota–gut–brain axis communication [74,75]. After stroke, several immunoregulatory pathways, such as the aryl hydrocarbon receptor (AHR) pathway, become activated, which is dependent on a dynamic balance between host-derived and GM-derived ligands (tryptophan derivatives as kynurenine and indoles), and also represent is a key regulatory pathway that mediates neuroinflammation [76]. Previous studies have shown that the activation of Microglia and AHR by host-derived ligands, represented by kynurenine, is detrimental after stroke, while the effects of post-stroke modification in AHR microbiota-derived ligands, by indoles, remain poorly understood [75]. However, AHR microbiota-derived ligands, such as indoles, are reduced in human plasma samples 24 h after ischemic stroke compared to controls, showing that post stroke GM changes could led to a loss of microbially-derived indole-based ligands of AHR [76]. In particular, tryptophan could be metabolized to tryptamine in a TrpD-dependent manner by bacteria belonging to the *Clostridium*, *Ruminococcus*, *Blautia*, and *Lactobacillus* genera [77]. Furthermore, tryptophan can be metabolized by the GM into indole and its derivatives [78] by Gram-negative and Gram-positive bacterial species, including *Escherichia coli* and *Clostridium* spp. [79]. Consistently, very low levels of indole and skatole were detected in our IS patient cohort, which might be related both to the GM imbalance and the decreased abundance of Ruminococcaceae and Clostridiales (Clostridiaceae).

Certainly, the small cohort sample size may represent a study limitation, leading to a need for a further scaleup of patients to confirm these findings.

## 5. Conclusions

The results obtained from this study suggest a possible relationship between IS and GM modifications. Moreover, an altered GM composition could have indirect effects on the production of metabolites involved in important brain functions, such as SCFAs, from an increase in microbial SCFA producers. Herein, specific fecal signatures characterizing IS patients have been uncovered, such as a high abundance of acetic acid and *Akkermansia,* coupled with a low abundance of the Clostridiales order and Ruminococcaceae family, which could represent group of bacteria and metabolite typically associated to ischemic stroke.

Finally, this study on a limited adult Italian population may represent a pilot study for later investigations on GM of IS patients. It may play a pivotal role for future larger studies on adults and for new translating the issue to the IS pediatric disease.

## Figures and Tables

**Figure 1 microorganisms-12-00037-f001:**
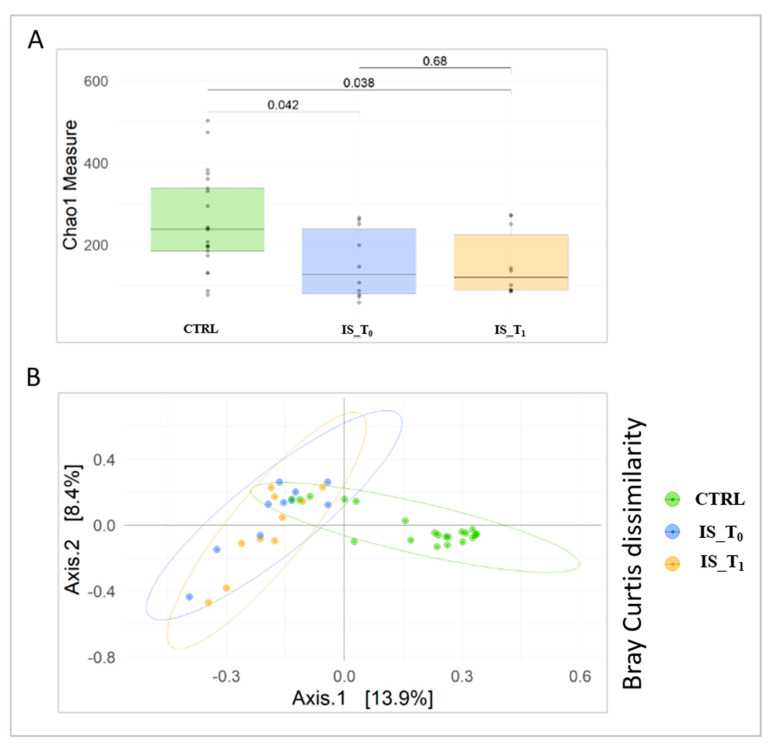
GM ecological analysis. (**A**) Alpha-diversity of IS T_0_ and IS T_1_ cohorts and CTRLs based on Chao-1 index (*p*-value by pairwise comparisons using the Mann-Whitney test). (**B**) Principal coordinate analysis (PCoA) plots shows beta diversity of IS T_0_ and IS T_1_ cohorts and CTRLs performed using the Bray-Curtis index. Legend: IS_T_0_: IS patients at T_0_; IS_T_1_: IS patients at T_1_; CTRL: controls subjects.

**Figure 2 microorganisms-12-00037-f002:**
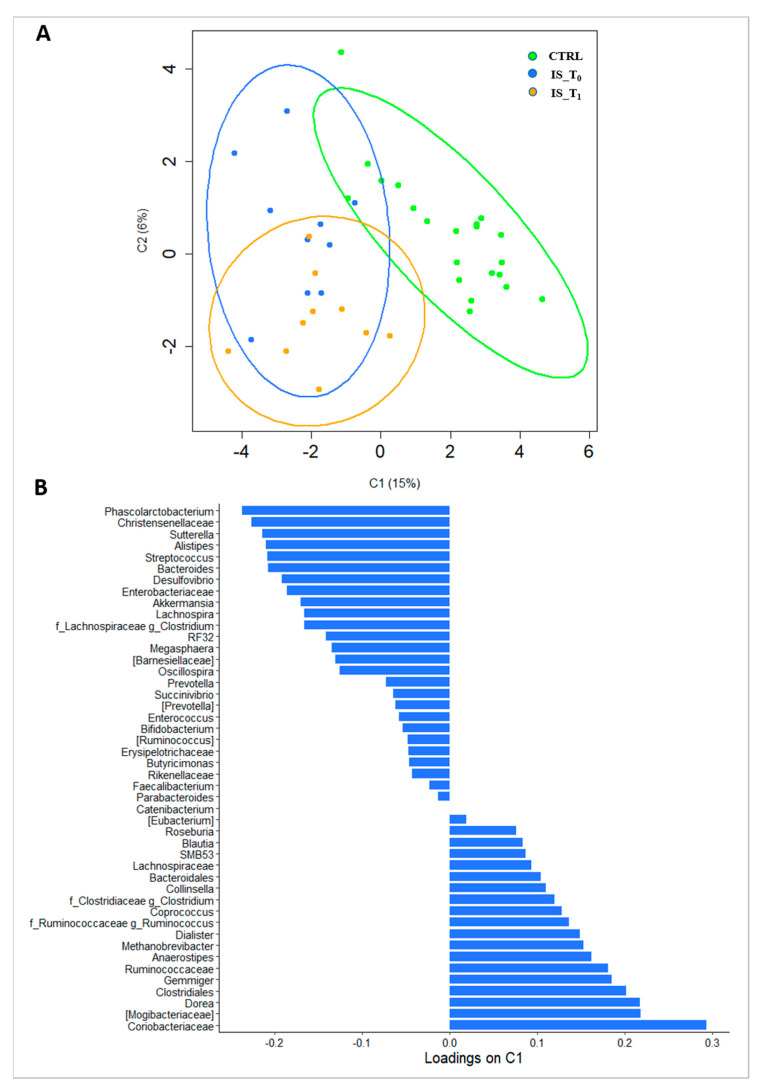
(**A**) Score plot of PLS-DA; (**B**) loading plot of all ASVs on Component 1. Legend: IS_T_0_: IS patients at T_0_; IS_T_1_: IS patients at T_1_; CTRL: controls subjects.

**Figure 3 microorganisms-12-00037-f003:**
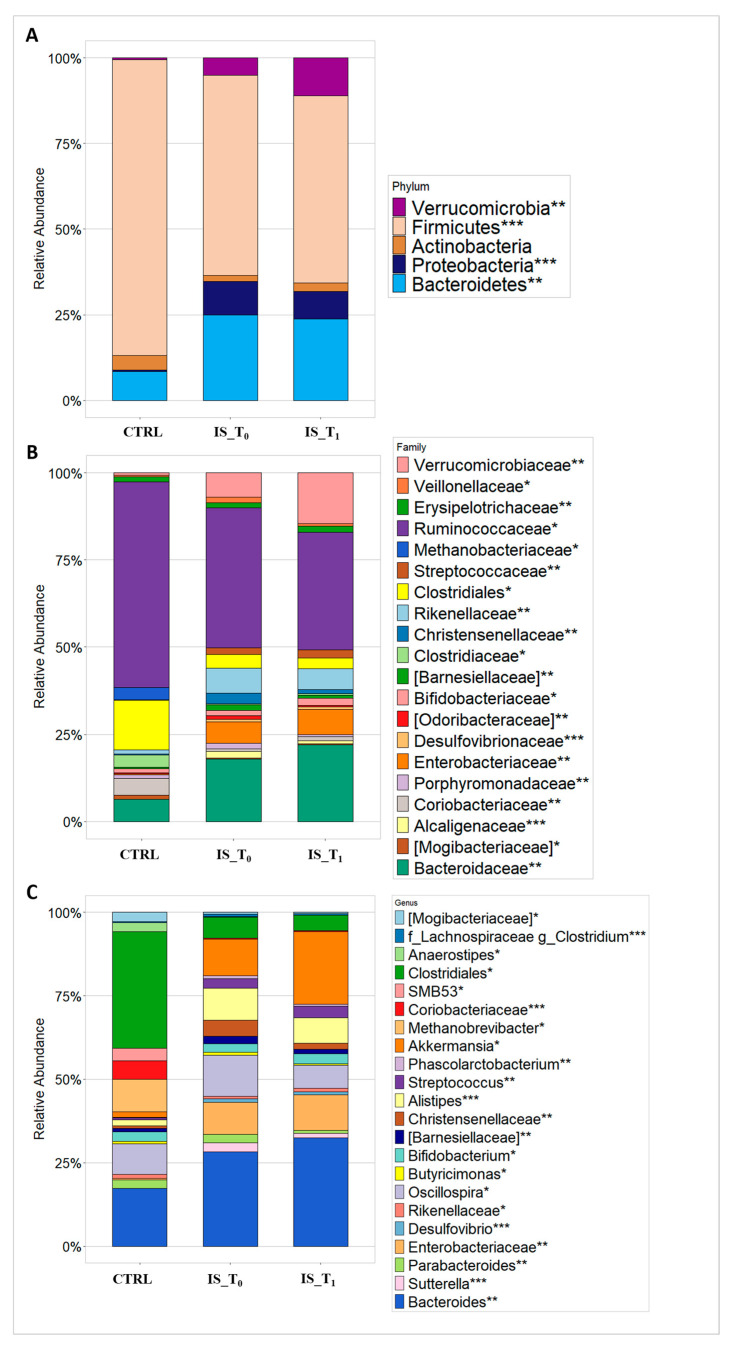
ASV distributions at L2 (**A**); L5 (**B**); and L6 (**C**) of the GM profiling of CTRLS, IS T_0_ and IS T_1_. At L5 and L6 ASV filtered by statistical significance based on the Kruskal–Wallis test. *** *p*-value FDR ≤ 0.001; ** *p*-value FDR ≤ 0.01; * *p*-value FDR ≤ 0.05. Legend: IS_T_0_: IS patients at T_0_; IS_T_1_: IS patients at T_1_; CTRL: controls subjects.

**Figure 4 microorganisms-12-00037-f004:**
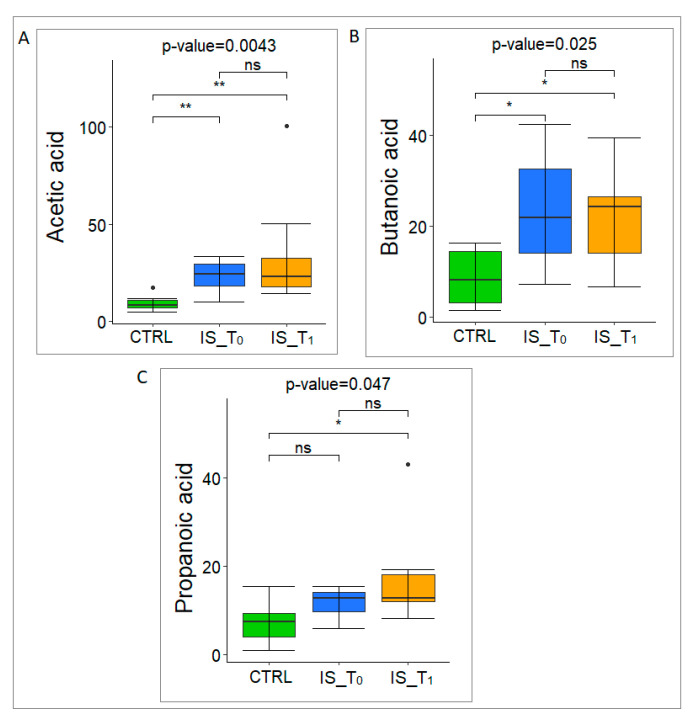
Distribution of SCFAs (Acetic Acid in panel A; Butanoic Acid in panel B; Propanoic Acid in panel C) in CTRL, IS patients at T_0_, IS patients at T_1_ (Kruskal–Wallis test, * *p*-value ≤ 0.05, ** *p*-value ≤ 0.01). IS_ T_0_: IS patients at T_0_; IS_T_1_: IS patients at T_1_; CTRL: controls subjects.

**Figure 5 microorganisms-12-00037-f005:**
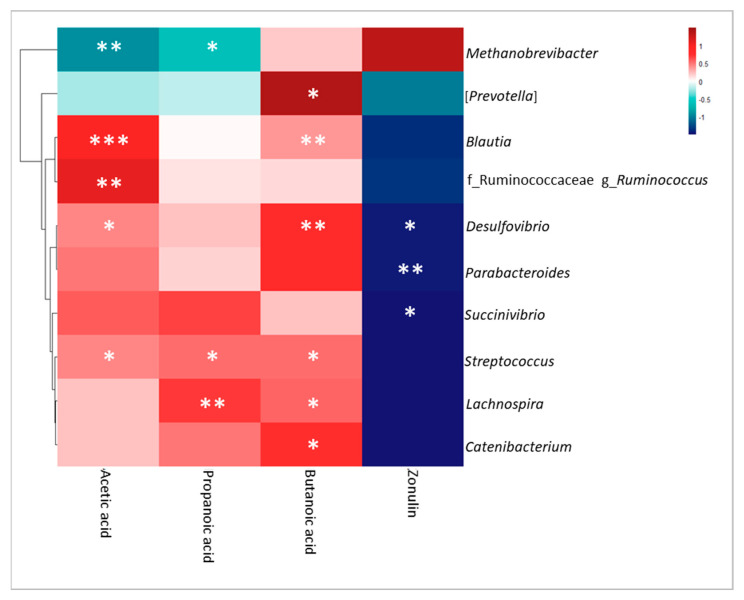
Correlation’s heatmap between ASVs, SCFAs, and zonulin performed using Ward’s clustering method and Euclidean distance.* *p*-value ≤ 0.05; ** *p*-value ≤ 0.01, *** *p*-value ≤ 0.001.

**Table 1 microorganisms-12-00037-t001:** Correlation coefficients between SCFAs, and zonulin with BBP related to IS patients at time T_0_.

Blood Biochemical Parameters (BBP)	Acetic Acid	Propanoic Acid	Butanoic Acid	Total SCFA	Zonulin
ALT (U/L)	−0.396	−0.463	−0.492	−0.543	0.402
AST (mU/ mL)	0.036	−0.064	−0.324	−0.19	−0.006
GGT (U/L)	−0.257	0.337	−0.238	−0.204	−0.198
Tryglicerides (mg/dL)	0.028	0.387	0.018	0.083	0.738 *
Cholesterol (mg/dL)	0.186	0.614	0.608	0.545	0.359
Glucose (mg/dL)	−0.152	0.556	0.033	0.04	0.568
Albumin (g/mL)	0.251	0.412	0.442	0.441	0.73 *
Bilirubin (mg/dL)	−0.516	−0.168	−0.255	−0.406	0.125

* Statistically significant (*p*-value ≤ 0.05).

## Data Availability

The datasets presented in this study can be found in online repositories. The names of the repository/repositories and accession number(s) can be found below: PRJNA1030992, PRJNA996768, PRJNA531579 (https://www.ncbi.nlm.nih.gov/bioproject).

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
