# Peer review of "Gut Microbiota Ecological and Functional Modulation in Post-Stroke Recovery Patients: An Italian Study"

_microorganisms, 2023, doi:10.3390/microorganisms12010037_

Round 1

Reviewer 1 Report

Comments and Suggestions for Authors

The manuscript is devoted to the gut microbiota taxonomy of the post-stroke recovery patients in association with some microbial metabolites. The design of the manuscript should be significantly revised.

1)      The small number of patients is an obvious limitation. However, the subsequent division of patients into subgroups looks very strange and controversial. Data on sample size calculation and power of the selected sample size were not demonstrated.

2)      Are the data from such a small number of patients distributed normally? Assuming not, then the data should be reported as median and IQR, but not as average or mean values with SD like in line 79.

3)      I suppose that the manuscript should consist of the data on the comparison of all patients at T0 and T1 and healthy controls without any subsequent analysis of “subdivided” groups, because it seems ridiculous to do any statistical analysis between 2 and 5 patients and so on. Otherwise, authors should include very compelling arguments and use specific statistical approaches that allow calculations to be made using very small sample sizes. However, it's still best to avoid this because the results will look inconclusive.

4)      The results on SCFA and other metabolites are presented as relative signals. There are lost of methods described in literature, which provides the quantitative analysis of the SCFA. Why was the GC/MS method used not validated? Quantitative data obtained using an unvalidated method appears unreliable.

5)      No pure chemical standards of the SCFAs were mentioned. Hence, the

6)      Hexanoic acid was used as an internal standard for the quantitative analysis of the SCFAs. However, this metabolite was detected in feces samples according to the Human Metabolome DataBase https://hmdb.ca/metabolites/HMDB0000535#concentrations. How did you use this acid as an internal standard if it was an endogenous fecal compound? This fact calls into question all the data presented on determining the level of SCFAs in the manuscript.

7)      It is best to use common terms, so gut microbiota ecology in this work is actually a gut microbiota taxonomy since 16S rRNA sequencing was used.

Author Response

Manuscript ID: 2713590

Gut microbiota ecological and functional modulation in post-stroke recovery patients

Reviewer #1: The manuscript is devoted to the gut microbiota taxonomy of the post-stroke recovery patients in association with some microbial metabolites. The design of the manuscript should be significantly revised.

AUTHORS’ REPLY.

We thank the Reviewer for the opportunity to improve our manuscript, which was revised to give more details as requested. We considered all suggestions indicated by the Reviewer and consistently we have reported the revisions along the new R1 text by using the “track change”.

We are aware that more number of patients is necessary to improve the robustness of the study but we want to give a piece of evidence on this disease as a pilot study that can be implemented with other patients cohorts.

The entire manuscript has been reviewed by a native English speaker and translator.

Major concerns

1)      The small number of patients is an obvious limitation. However, the subsequent division of patients into subgroups looks very strange and controversial. Data on sample size calculation and power of the selected sample size were not demonstrated.

AUTHORS’ REPLY.

Thanks the reviewer for the observation. We know that our patients cohort and the following sample sizes is small. Regarding subgroup, in which statistical power is lacking, are created in order to consider the hypothetical role of the main anamnestic and clinical variables. In addition, as you suggest for these kind of omics studies the dimension of sample size is not demonstrated as reported into international literature.

In order to better describe that the samples size is limitation of this study we introduced in the Discussion section, the sentence as follows:

“Certainly, the small cohort sample size, and also for stratified groups, may represent a study limitation, and it will be assessed a further scaleup of patients’ set.”

2)      Are the data from such a small number of patients distributed normally? Assuming not, then the data should be reported as median and IQR, but not as average or mean values with SD like in line 79.

AUTHORS’ REPLY.

Thanks to the Reviewer for this comment, we have calculated and reported the median age, as suggested, in Materials and Methods in “2.1 Study participants and Clinical data” as follows:

“Ten patients with ischemic stroke (IS) aged 54-81 years (median, 70.5 years ± IQR 14) of whom 6 males and 4 females, were recruited at the Villa Beretta Rehabilitation Center, Valduce Hospital, (Costa Masnaga, Lecco, Italy) between January and December 2021. Six aged-matched healthy subjects as adults controls (CTRLs) (median, 44.5 years ± IQR 30.25)”

3)      I suppose that the manuscript should consist of the data on the comparison of all patients at T0 and T1 and healthy controls without any subsequent analysis of “subdivided” groups, because it seems ridiculous to do any statistical analysis between 2 and 5 patients and so on. Otherwise, authors should include very compelling arguments and use specific statistical approaches that allow calculations to be made using very small sample sizes. However, it's still best to avoid this because the results will look inconclusive.

AUTHORS’ REPLY.

Thanks the reviewer for the observation. We know that the stratification of our cohort into subgroups may have reduced the statistical power of the results and may represent a limitation of the study. However, aware of the limitation due to the small sample size, we still stratified the samples in order to consider the hypothetical role of the main anamnestic most relevant clinical variables for this pathology (as motricity index and walking tests) in order to provide a preliminary indication of their potential influence on the disease. Moreover, given the low number and the lack of robustness of the analysis we focused discussion and conclusions only on differences between Is patients and CTRLs without mentioning the role of variables in depth.

In order to better describe that the samples size is limitation of this study we introduced in the Discussion section, the sentence as follows:

“Certainly, the small cohort sample size, and also for stratified groups, may represent a study limitation, and it will be assessed a further scaleup of patients’ set.”

4)      The results on SCFA and other metabolites are presented as relative signals. There are lost of methods described in literature, which provides the quantitative analysis of the SCFA. Why was the GC/MS method used not validated? Quantitative data obtained using an unvalidated method appears unreliable.

AUTHORS’ REPLY.

Thanks the reviewer for the comment.

Firstly we missing the relative reference of extraction and processing of samples to detect SCFAs in order to describe the methods adopted.

Now, we introduced the reference in Materials and Methods in “2.5 Gas Chromatography coupled to Mass Spectrometry (GC-MS) Metabolomics” as follows:

“Finally, 1 µl of each sample was injected into GC-MS. The temperature program was: 50°C for 2 minutes, a temperature increase of 10°C per minute at 70°C, 3°C per minute at 85°C, 5°C per minute at 110°C, 15°C per minute at 200°C, 20°C per minute at 230°C, 10°C per minute at 240°C, then 240°C for 5 minutes according to the revised procedure performed by Zhao et al [21].”

Moreover, the large part of literature in terms of omics study, by GC-MS, showed methods similar to ours, in which no validation was performed and requested.

Please see these references:

- Ziemons J, Aarnoutse R, Heuft A, Hillege L, Waelen J, de Vos-Geelen J, Valkenburg-van Iersel L, van Hellemond IEG, Creemers GM, Baars A, Vestjens JHMJ, Penders J, Venema K, Smidt ML. Fecal levels of SCFA and BCFA during capecitabine in patients with metastatic or unresectable colorectal cancer. Clin Exp Med. 2023 Nov;23(7):3919-3933. doi: 10.1007/s10238-023-01048-7. Epub 2023 Apr 7. PMID: 37027066; PMCID: PMC10618330.

- Wang Q, Chen C, Zuo S, Cao K, Li H. Integrative analysis of the gut microbiota and faecal and serum short-chain fatty acids and tryptophan metabolites in patients with cirrhosis and hepatic encephalopathy. J Transl Med. 2023 Jun 17;21(1):395. doi: 10.1186/s12967-023-04262-9. PMID: 37330571; PMCID: PMC10276405.

- Igudesman D, Crandell JL, Corbin KD, Hooper J, Thomas JM, Bulik CM, Pence BW, Pratley RE, Kosorok MR, Maahs DM, Carroll IM, Mayer-Davis EJ. Associations of Dietary Intake with the Intestinal Microbiota and Short-Chain Fatty Acids Among Young Adults with Type 1 Diabetes and Overweight or Obesity. J Nutr. 2023 Apr;153(4):1178-1188. doi: 10.1016/j.tjnut.2022.12.017. Epub 2022 Dec 27. PMID: 36841667; PMCID: PMC10356993.

5)      No pure chemical standards of the SCFAs were mentioned. Hence, the

AUTHORS’ REPLY.

Thanks to the Reviewer for the observation. Firstly, before samples analysis, we performed many experiments using pure chemical standards of SCFAs alone or in mixture in order to evaluate retention times, peaks form and compounds identification. Moreover, we prepared spikes at different concentrations to avoid matrix effects.

Now, we introduced the reference in Materials and Methods in “2.5 Gas Chromatography coupled to Mass Spectrometry (GC-MS) Metabolomics” as follows:

“ 2.5    Gas Chromatography coupled to Mass Spectrometry (GC-MS) Metabolomics Analysis

SCFAs (A), indole and 3-methyl-indole (skatole) (B) concentration were estimated by Gas Chromatograph (GC) 7890A (Agilent Technologies, Santa Clara, CA, USA) coupled to a mass spectrometer (MS) 5977 (Agilent Technologies, Santa Clara, CA, USA) in electron impact mode (ionization voltage of 70 eV) equipped with a DB-HeavyWAX (60m x 0.250 mm x 0.25 µm) capillary column Agilent Technologies.

  • An aqueous stock standard solution was prepared from chemical reagents of acetic acid (99%), propanoic acid (99%) and butanoic acid (99%) purchased by Merck (Darmstadt, Germany) with a concentration of 200 mm for each acid. All the stock standard solutions were stored at −20°C until used.
  • For SCFAs analysis, fecal samples (250 mg) were added with 1.25 mL of H2O and homogenized on a shaker for 3 minutes. Then the samples were acidified with HCl (3M) to a pH of 2-3. Finally, they were centrifuged at 5000g at 4°C for 10 min. After centrifugation 1 ml of supernatant was centrifuged at 15000g at 4°C for 15 min. The supernatant was transferred and basified at pH 6 with NaOH (5M), then was added with hexanoic acid (final concentration of 0.4 mg/ml) as an internal standard (before sample screening). The samples were filtered through 0.45 um and 0.22 um filters. Finally, 1 µl of each sample was injected into GC-MS. The temperature program was: 50°C for 2 minutes, a temperature increase of 10°C per minute at 70°C, 3°C per mi-nute at 85°C, 5°C per minute at 110°C, 15°C per minute at 200°C, 20°C per minute at 230°C, 10°C per minute at 240°C, then 240°C for 5 minutes according to the adopted in house procedures performed by Zhao et al [21].
  • For indole and 3-Methyl-Indole determinations, fecal samples (100-500 mg) were managed and analysed according to the procedure performed by Vernocchi et al. [22] by using GC-MS coupled to Solid Phase Micro-Extraction (SPME).

The compounds identification was confirmed by injection of pure standards and a comparison of the retention time and corresponding MS spectra. Moreover, as a support the identification of all molecules was conducted searching mass spectra in the as National Institute of Standards and Technology (NIST, Rockville, MD, version 2d, build April 26, 2005) library and in the literature. Moreover, the chromatograms were integrated and the quantitative data, expressed in mg/kg, were obtained by interpolation of the relative areas than the area of the internal standard..”

6)      Hexanoic acid was used as an internal standard for the quantitative analysis of the SCFAs. However, this metabolite was detected in feces samples according to the Human Metabolome DataBase https://hmdb.ca/metabolites/HMDB0000535#concentrations. How did you use this acid as an internal standard if it was an endogenous fecal compound? This fact calls into question all the data presented on determining the level of SCFAs in the manuscript.

AUTHORS’ REPLY.

Thanks to the Reviewer for the comment. Firstly, we have investigate the presence of the 3 principal SCFAs as: acetic, butyric and propionic acids according to dedicated methods published from Zhao et al. (Zhao G, Nyman M, Jönsson JA. Rapid determination of short-chain fatty acids in colonic contents and faeces of humans and rats by acidified water-extraction and direct-injection gas chromatography. Biomed Chromatogr. 2006 Aug;20(8):674-82. doi: 10.1002/bmc.580. PMID: 16206138.).

In addition, we know that the typical internal standard for this analysis was 2-Ethylbutyric acid, unfortunately it was not available at that time.

Hence, we know that hexanoic acid was potentially present but our focus was different (only on the 3 SCFAs), therefore we made a preliminary screening on the 16 samples in order to explore samples for the presence of Hexanoic acid.

It was not present in any sample and therefore we proceeded to introduce hexanoic acid as an internal standard.

Moreover, we we modified sentence as follows in Materials and Methods section “Gas Chromatography coupled to Mass Spectrometry (GC-MS) Metabolomics Analysis”:

“For SCFAs analysis, fecal samples (250 mg) were added with 1.25 mL of H2O and homogenized on a shaker for 3 minutes. Then the samples were acidified with HCl (3M) to a pH of 2-3. Finally, they were centrifuged at 5000g at 4°C for 10 min. After centrifugation 1 ml of supernatant was centrifuged at 15000g at 4°C for 15 min. The supernatant was transferred and basified at pH 6 with NaOH (5M), then was added with hexanoic acid (final concentration of 0.4 mg/ml) as an internal standard (before sample screening). The samples were filtered through 0.45 um and 0.22 um filters. Finally, 1 µl of each sample was injected into GC-MS. The temperature program was: 50°C for 2 minutes, a temperature increase of 10°C per minute at 70°C, 3°C per minute at 85°C, 5°C per minute at 110°C, 15°C per minute at 200°C, 20°C per minute at 230°C, 10°C per minute at 240°C, then 240°C for 5 minutes according to the adopted in house procedures performed by Zhao et al [21]

Regarding indole and 3-methyl indole we used the methods performed by Vernocchi et al (Microorganisms 2020, 8, 1540, doi:10.3390/microorganisms8101540) used for several of our pubblications, in which we compared peaks with NIST library (Liver Int. 2021 Jun;41(6):1320-1334. doi: 10.1111/liv.14876. Epub 2021 Mar 25. PMID: 33713524; Int J Mol Sci. 2020 Nov 19;21(22):8730. doi: 10.3390/ijms21228730; Microorganisms. 2020 Oct 6;8(10):1540. doi: 10.3390/microorganisms8101540. PMID: 33036309; J Transl Med. 2020 Feb 3;18(1):49. doi: 10.1186/s12967-020-02231-0)

7)      It is best to use common terms, so gut microbiota ecology in this work is actually a gut microbiota taxonomy since 16S rRNA sequencing was used.

AUTHORS’ REPLY.

Thanks to the Reviewer for this suggestion, we replaced “gut microbiota ecology” with “gut microbiota taxonomy” in all manuscript.

Reviewer 2 Report

Comments and Suggestions for Authors

The matierals and methods in abstract us very poor must be reorganized with more information 

The conclusion part in the abstract must be reordered to b more clear. ..

The keywords must be written in capital first letters 

In introduction there were many important references not cited please check the references number 3 and 4 ....from line 55 till 67 please this part should be reorganized. ..

In materials where is the ethical committee form in addtion where the study inclusion and exclusion criteria. ... .how could authors calculate the dose please mention this however the sample size is too small to be calculated... in results all paragraphs should be written in clear detailed form because many lines are so confused.....discussion and conclusion are ok 

Comments on the Quality of English Language

Need English editing

Author Response

Manuscript ID: 2713590

Gut microbiota ecological and functional modulation in post-stroke recovery patients

Reviewer #2

We thank the Reviewer for the opportunity to improve our manuscript, which was revised to give more details as requested. We considered all suggestions indicated by the Reviewer and consistently we have reported the revisions. The entire manuscript has been reviewed by a native English speaker and translator.

Major concerns

  1. The materials and methods in abstract us very poor must be reorganized with more information

AUTHORS’ REPLY.

We thank the Reviewer for this advice. We have modified the abstract and added more information, reported as follows:

Abstract: Ischemic stroke (IS) can be caused by perturbation of the gut-brain axis. An imbalance in the gut microbiota (GM) or dysbiosis, may be linked to several IS risk factors and could influence the brain producing different metabolites such as short chain fatty acids (SCFAs), indole derivatives. This study examine GM ecological changes and its metabolic activities after stroke. Fecal samples of 10 IS patients compared to 6 healthy subjects controls (CTRLs) have been evaluated in terms of ecological profile by 16S rRNA taxonomy and functional profile by metabolomics analysis performed with gas chromatograph coupled to a mass spectrometer (GC-MS), while marker of gut permeability, the zonulin, was detected by enzyme-linked immuno assay (ELISA) test. Data were analyzed by univariate and multivariate statistical analysis and correlated with clinical features and biochemical variables by correlation and nonparametric tests. Particularly, the results revealed the high concentration of metabolites involved in important brain functions such as SCFAs by also detecting the producing species, as Blautia, Akkermansia. On the contrary, indole and 3-methyl indole decreased compared to CTRLs. In conclusion, this study highlighted as IS could change gut microbial environments through indication of specific microbial as Akkermansia and metabolic biomarkers as acetic acid that characterizing IS patients and could be considered potential disease phenotype signatures.

  1. The conclusion part in the abstract must be reordered to b more clear. ..

AUTHORS’ REPLY.

We thank the Reviewer for this advice. We have modified the conclusion as follows: “In conclusion, this study highlighted as IS could change gut microbial environments through indication of specific microbial as Akkermansia and metabolic biomarkers as acetic acid that characterizing IS patients and could be considered potential disease phenotype signatures

  1. The keywords must be written in capital first letters

AUTHORS’ REPLY.

We thank the Reviewer for this advice. We have written keywords in capital first letters in the manuscript.

  1. In introduction there were many important references not cited please check the references number 3 and 4 ....from line 55 till 67 please this part should be reorganized.

AUTHORS’ REPLY.

We thank the Reviewer for this suggestion and we have introduced some new references as follows:

 [4] Bercik, P.; Collins, S.M.; Verdu, E.F. Microbes and the Gut‐brain Axis. Neurogastroenterology Motil 2012, 24, 405–413, doi:10.1111/j.1365-2982.2012.01906.x.

[5] Li, X.; You, X.; Wang, C.; Li, X.; Sheng, Y.; Zhuang, P.; Zhang, Y. Bidirectional Brain‐gut‐microbiota Axis in Increased Intestinal Permeability Induced by Central Nervous System Injury. CNS Neurosci Ther 2020, 26, 783–790, doi:10.1111/cns.13401.

[6] Yarandi, S.S.; Peterson, D.A.; Treisman, G.J.; Moran, T.H.; Pasricha, P.J. Modulatory Effects of Gut Microbiota on the Central Nervous System: How Gut Could Play a Role in Neuropsychiatric Health and Diseases. J Neurogastroenterol Motil 2016, 22, 201–212, doi:10.5056/jnm15146.

[7] Heiss, C.N.; Olofsson, L.E. The Role of the Gut Microbiota in Development, Function and Disorders of the Central Nervous System and the Enteric Nervous System. J Neuroendocrinology 2019, 31, e12684, doi:10.1111/jne.12684.

The paragraph between line 55 and 67 has been changed as follows:

Under normal circumstances, the GM performs many crucial physiological functions, such as the maintenance of gut integrity [9], the production of SCFAs via fermentation of complex polysaccharides,  the biosynthesis of essential amino acids and vitamins [10], and protection against pathogens by regulating host immunity [11]. On the other hand, an imbalance within the GM (i.e., dysbiosis) is related to numerous GI and metabolic diseases, including diabetes, dyslipidemia and obesity [12] which, along with hypertension and atherosclerosis, represent important risk factors for ischemic stroke (IS).

IS occurs when an obstruction leads to substantial decrease of blood flow to the brain [13]. Furthermore, IS can lead to alterations in the composition of the GM and its functions, and, conversely, GM alterations and changes in GM-derived metabolites have been associated with IS risk [14]. Several studies have linked patient prognosis following IS with GM dysbiosis [15] and altered SCFA production, which in turn have numerous homeostatic and anti-inflammatory effects related to GM biochemistry [16].

IS can occur at any age or stage of life. While the incidence of pediatric stroke is considerably lower than it is in adults, the neurological consequences [17] showing an impact on life quality and on the health care service is significantly higher following pediatric stroke [18]

  1. In materials where is the ethical committee form in addition where the study inclusion and exclusion criteria. ... .how could authors calculate the dose please mention this however the sample size is too small to be calculated

AUTHORS’ REPLY.

We thank the Reviewer for this observation. If we have interpreted your question, here are the answers.

For probiotics: we exclude anyone who takes a probiotic (as supplement or food) regardless of concentration.

For the antibiotics: as antibiotics can affect the gut ecology, we exclude all patients who takes antibiotics independently of dose.

  1. In results all paragraphs should be written in clear detailed form because many lines are so confused.....discussion and conclusion are ok

AUTHORS’ REPLY.

We thank the Reviewer for this suggestion.  We have revised the entire Results section to give it more readable and understandable.

Reviewer 3 Report

Comments and Suggestions for Authors

The manuscript entitled: Gut microbiota ecological and functional modulation in post-2 stroke recovery patients, addresses an interesting research point in using 16S rRNA and functional profile by metabolomics analysis in link with ischemic stroke (IS). The manuscript focuses on an interesting topic of the bidirectional gut-brain axis using up to date approaches such as microbiome and metabolome. However, the methodologies are not fully described to be easy to be followed by the readers. In addition, the low number of subjects that Have included in this investigation is a limiting factor.

I recommend major revision of this manuscript entitled: Gut microbiota ecological and functional modulation in post-2 stroke recovery patients, I wonder if the authors could include data from subjects that have similar age or those available from open sources.  The evaluation is based on that overall value of data presented and novelty of the idea. In addition to, the scope of the topic the manuscript addressed.

Author Response

Evaluation Report

Manuscript Title:

Gut microbiota ecological and functional modulation in post-2 stroke recovery patients

Journal

Micoorganisms

Manuscript ID

Micoorganisms -2713590

Section

Comments and recommendation

General comment

The manuscript entitled: Gut microbiota ecological and functional modulation in post-2 stroke recovery patients, addresses an interesting research point in using 16S rRNA and functional profile by metabolomics analysis in link with ischemic stroke (IS). The manuscript focuses on an interesting topic of the bidirectional gut-brain axis using up to date approaches such as microbiome and metabolome. However, the methodologies are not fully described to be easy to be followed by the readers. In addition, the low number of subjects that Have included in this investigation

is a limiting factor.

AUTHORS’ REPLY.

We thank the Reviewer for the opportunity to improve our manuscript, which was revised to give more details as requested. We considered all suggestions indicated by the Reviewer and consistently we have reported the revisions.

The entire manuscript has been reviewed by a native English speaker and translator.

Abstract

1- The abstract is well structured and describe well all contents of the study. The objective describes well the purpose of the study

2- However, the authors the conclusion is so general and it

needs to be more specific to the findings of the study.

AUTHORS’ REPLY.

We thank the Reviewer for this advice. We have modified all abstract and we also made the conclusions clearer as follows: “Ischemic stroke (IS) can be caused by perturbation of the gut-brain axis. An imbalance in the gut microbiota (GM) or dysbiosis, may be linked to several IS risk factors and could influence the brain producing different metabolites such as short chain fatty acids (SCFAs), indole derivatives. This study examine GM ecological changes and its metabolic activities after stroke. Fecal samples of 10 IS patients compared to 6 healthy subjects controls (CTRLs) have been evaluated in terms of ecological profile by 16S rRNA taxonomy and functional profile by metabolomics analysis performed with gas chromatograph coupled to a mass spectrometer (GC-MS), while marker of gut permeability, the zonulin, was detected by enzyme-linked immuno assay (ELISA) test. Data were analyzed by univariate and multivariate statistical analysis and correlated with clinical features and biochemical variables by correlation and non-parametric tests. Particularly, the results revealed the high concentration of metabolites involved in important brain functions such as SCFAs by also detecting the producing species, as Blautia, Akkermansia. On the contrary, indole and 3-methyl indole decreased compared to CTRLs. In conclusion, this study highlighted as IS could change gut microbial environments through indication of specific microbial as Akkermansia and metabolic biomarkers as acetic acid that characterizing IS patients and could be considered potential disease phenotype signatures. “

Introduction

1- The introduction section is well written and described the well the state of the art of study problem. The authors used up to date research articles relevant to the study topic.

AUTHORS’ REPLY.

Thank you very much to the Reviewer.

Materials and methods

The experiment of this study is well designed and is fitting with the objectives and the research methodology utilized relevant and properly administered. In addition, methods used for statistical data analysis is suitable for these types of data.

AUTHORS’ REPLY.

Thank you very much to the Reviewer.

2- However, the low number of subjects that Have included in this investigation is a limiting factor.

AUTHORS’ REPLY.

Thanks the reviewer for the observation. We know that the subjects’ cohort and the sample sizes obtained by stratification into subgroups to consider clinical features, could reduce the statistical power of the results and represent a limitation of the study. Indeed we have add a sentence in Discussion section as follows:

“Certainly, the small cohort sample size, and also for stratified groups, may represent a study limitation, and it will be assessed a further scale up of patients’ set.”

3- The average age difference between the ischemic stroke group (68.9 years) and healthy controls subjects (44.16 years) is another second limiting factor as well established that age is a crucial factor that affect microbiome profile.

AUTHORS’ REPLY.

Thanks the reviewer for the comment. We replaced the mean with the median. Now we introduced the median values instead of mean values in order to consider the non-normal distribution of samples.

Furthermore, we realized the difference in age and implemented a test of confounding values ​​to make sure that age was not confounding and that there was no statistical significance between the two groups (beta diversity not significant).

We modified words in Materials and Methods in “2.1 Study participants and Clinical data” as follows:

“Ten patients with ischemic stroke (IS) aged 54-81 years (median, 70.5 years ± IQR 14) of whom 6 males and 4 females, were recruited at the Villa Beretta Rehabilitation Center, Valduce Hospital, (Costa Masnaga, Lecco, Italy) between January and December 2021. Six aged-matched healthy subjects as adults controls (CTRLs) (median, 44.5 years ± IQR 30.25) were recruited during an epidemiological survey executed at the Human Microbiome Unit of Bambino Gesù Children’s Hospital in Rome (BBMRI Human Micro-biome Biobank, OPBG) to produce a digital biobank of healthy subjects (controls, CTRLs).”

4- The methodologies are not fully described to be easy to be followed by the readers. For example, the authors mentioned Zonulin, was detected with ELISA kit (Materilas and methods section line 167) without giving all detailed

information of this kit.

AUTHORS’ REPLY

Thanks to the Reviewer for this comment, we have improved and implemented Materials and Methods section, particularly for Zonulin section, has been fully described as follows:

“Zonulin, was detected with ELISA kit (IDK Zonulin ELISA for the in vitro determination of zonulin family peptides, ZFP, in stool, Immunodiagnostik AG, Bensheim, Germa-ny) only for the IS patients  at T0 and T1 time points according to the manufacturer’s protocol using 15 mg of patient stool sample. Firstly, a biotinylated zonulin tracer was added to samples, standards, and controls. For an hour, the treated samples, standards, and controls were incubated in microtiter plates coated with an-ti-zonulin polyclonal antibodies in the wells of microtiter plates. Samples were then washed five times in 250 µL wash buffer, and the remaining liquid was absorbed. The biotinylated zonulin tracer is associated with peroxidase and can be found in streptavidin (100 µL), which was subsequently added to each well. The microplate was incubated, washed, and drained as described previously. For 15 minutes, the samples were incubated with 100 µL substrate solution, and then the reaction was brought to an end with an acidic stop solution. The absorption was determined immediately at 450 nm against a reference wave of 620 nm using Infinite® F50 compact ELISA absorbance microplate reader (TECAN, Männedorf, Switzer-land). The quantitative data were represented by zonulin concentrations in ng/ml.”

Results & Discussion

1- The results and discussion of this manuscript are described the findings of the manuscript. The results are well discussed in the discussion section using up to date

literature which closely related to the study topic.

AUTHORS’ REPLY.

Thank you very much to the Reviewer.

Conclusions

The conclusions of this manuscript described well the

manuscript findings.

AUTHORS’ REPLY.

Thank you very much to the Reviewer.

Bibliography/References

The list of references is well formatted according to journal instructions. Most of references listed in the manuscript is up to date (2015-2021) and closely adhered to the research focus of study although few are not recent (1981,

1985, 1995, 2002, and 2005).

AUTHORS’ REPLY.

Thanks the reviewer for this observation. The cited references are related to the motricity indexes and included the normative data used in current clinical practice and constitute the reference database. Hence, the clinicians use them commonly and there are, no more recent normative data.

Recommendation and final comment

I recommend major revision of this manuscript entitled: Gut microbiota ecological and functional modulation in post-2 stroke recovery patients, I wonder if the authors could include data from subjects that have similar age or those available from open sources. The evaluation is based on

that overall value of data presented and novelty of the idea.

AUTHORS’ REPLY.

Thank you to the Reviewer for these suggestions but it is very difficult to implement our sample cohort for several reasons. Firstly, different analytical methods of the samples, which make the data not comparable. Furthermore, the microbiota is affected from several factors including ethnicity, geographical origin, eating habits, so it is very difficult to make consistent comparisons.

In addition to, the scope of the topic the manuscript

addressed.

AUTHORS’ REPLY.

Thank you very much to the Reviewer.

Round 2

Reviewer 1 Report

Comments and Suggestions for Authors

Despite the apparent significant change in the article, it is mainly associated with proofreading the English, and not with a semantic change in the content of the article. I still do not consider it appropriate to divide the already small number of patients into subgroups. Thus, I believe that the article should not be published as long as information about subgroups is present in the article in any form.
Regarding the information that has appeared on the use of conditions for determining SCFAs with reference to publications from the literature. The results of the protocols of different authors may not be reproduced among themselves. The absence of the method validation by your own team on your equipment under your conditions can lead to unknown measurement errors. Thus, I believe that the accuracy of your results is still questionable.

Author Response

Manuscript ID: 2713590

Gut microbiota ecological and functional modulation in post-stroke recovery patients

Reviewer #1: Despite the apparent significant change in the article, it is mainly associated with proofreading the English, and not with a semantic change in the content of the article. I still do not consider it appropriate to divide the already small number of patients into subgroups. Thus, I believe that the article should not be published as long as information about subgroups is present in the article in any form.

AUTHORS’ REPLY.

We thank the Reviewer for the further opportunity to improve our manuscript, which has been revised at this stage following the more suggestions indicated by the Reviewer 1.

As suggested by the Reviewer 1, the already small number of patients has not been divided into subgroups and, for this reason, subgroups have been removed in the current R2 edition.

However, we have maintained as clinical feature only the continuous variable MI_TOT (“Motricity Index test”) which did not request any patients’ regrouping. Therefore, with the current requested approach the GM analyses have been focused only on the two major group IS patients (T0 and T1) versus CTRLs. Moreover, in the current manuscript, we have modified some analyses by including new CTRLs dataset (n=15) age matched, as suggested to Reviewer 3.

Regarding the information that has appeared on the use of conditions for determining SCFAs with reference to publications from the literature. The results of the protocols of different authors may not be reproduced among themselves. The absence of the method validation by your own team on your equipment under your conditions can lead to unknown measurement errors. Thus, I believe that the accuracy of your results is still questionable.

AUTHORS’ REPLY.

We agree with the Reviewer 1 concerning the need to include a validation method for our assay. Based on this last observation, we have carried out in house specific validation of the SCFA determination method. Particularly, we made a calibration curve by considering acetic, propanoic and butanoic acid in different concentration range (see in the Table 1 below).

The linearity of the calibration curve for each SCFA was assessed by the calculation of the coefficient of determination (R2). Validation  was based on coefficient of variability (CV) of inter- and intra-day reproducibility (%), limit of detection (LOD), limit of quantification (LOQ) and % of recovery (by considering a point in the middle of the curve), according to European Medicines Agency (EMEA) and FDA guidelines (European Medicines Agency. ICH Guideline M10 on Bioanalytical Method Validation Step 2b. 2019. Available online: https://www.ema.europa.eu/en/documents/scientific-guideline/draft-ich-guideline-m10-bioanalytical-method-validationstep-2b_en.pdf (accessed on 3 April 2022); U.S. Department of Health and Human Services; Food and Drug Administration; Center for Drug Evaluation and Research (CDER); Center for Veterinary Medicine (CVM). Bioanalytical Method Validation—Guidance for Industry. 2018. Available online: http://www.fda.gov/downloads/Drugs/Guidances/ucm070107.pdf (accessed on 3 April 2022).) Moreover, we have also processed the samples in order to confirm the reproducibility of analysis.

Hence, in the Methods section “2.5  Gas Chromatography coupled to Mass Spectrometry (GC-MS) Metabolomics Analysis “ we added a sentence as follows:

“For SCFAs analysis, fecal samples (250 mg) were added to 1.25 mL of H2O and homogenized on a shaker for 3 minutes. The samples were then acidified with HCl (3M) to a pH of 2-3. Finally, they were centrifuged at 5000rpm at 4°C for 10 min. After centrifugation 1 ml of supernatant was centrifuged at 15000rpm at 4°C for 15 min. The supernatant was transferred and basified at pH 6 with NaOH (5M), after which hexanoic acid (final concentration of 0.4 mg/ml) was added as an internal standard before sample screening. The samples were then filtered through 0.45 µm and 0.22 µm filters. Finally, 1 µl of each sample was injected into GC-MS. The temperature program was: 50°C for 2 minutes, a temperature increase of 10°C per minute at 70°C, 3°C per minute at 85°C, 5°C per minute at 110°C, 15°C per minute at 200°C, 20°C per minute at 230°C, 10°C per minute at 240°C, then 240°C for 5 minutes, according to the adapted in house procedures performed by Zhao et al [25]. The method was validated by in house method according to European Medicines Agency (EMEA) and FDA guidelines (European Medicines Agency. ICH Guideline M10 on Bioanalytical Method Validation Step 2b. 2019. Available online: https://www.ema.europa.eu/en/documents/scientific-guideline/draft-ich-guideline-m10-bioanalytical-method-validationstep-2b_en.pdf (accessed on 3 April 2022); U.S. Department of Health and Human Services; Food and Drug Administration; Center for Drug Evaluation and Research (CDER); Center for Veterinary Medicine (CVM). Bioanalytical Method Validation—Guidance for Industry. 2018. Available online: http://www.fda.gov/downloads/Drugs/Guidances/ucm070107.pdf (accessed on 3 April 2022)) on bioanalytical method validation. The linearity of the calibration curve for each SCFA was assessed by the calculation of the coefficient of determination (R2). Validation was based on coefficient of variability (CV) of inter- and intra-day reproducibility (%), limit of detection (LOD), limit of quantification (LOQ) and % of recovery.”

Table 1. Overview of validation parameters

Analyte

Calibration range (mg/kg)

aLOD (mg/kg)

bLOQ (mg/kg)

Acetic acid

1-200

0.015

0.21

Propanoic acid

1-100

0.022

0.1

Butanoic acid

1-100

0.035

0.11

Analyte

R2

cCV Inter-day Precision (%)

CV Intra-day Precision (%)

Acetic acid

0.99±0.02

8

6

Propanoic acid

0.98±0.02

10

8

Butanoic acid

0.98±0.01

12

5

Analyte

Recovery (%)

Acetic acid

98

100  mg/kg

Propanoic acid

95

50  mg/kg

Burytic acid

50 mg/kg

105

aLOD : limit of detection; bLOQ: limit of quantification;

cCV: coefficient of variability

Reviewer 3 Report

Comments and Suggestions for Authors

The authors tried hard to improve the content of this manuscript. However, they did not include any additional data or even cannot justify the average age difference between the ischemic stroke group (68.9 years) and healthy controls subjects (44.16 years) which is considered a key factor affecting mirobiome profile. In this regard, i ask for major revision including more data as recommended from the  first revision.  

Author Response

Manuscript ID: 2713590, R2 REVISION

Gut microbiota ecological and functional modulation in post-stroke recovery patients

Reviewer #3: The authors tried hard to improve the content of this manuscript. However, they did not include any additional data or even cannot justify the average age difference between the ischemic stroke group (68.9 years) and healthy controls subjects (44.16 years) which is considered a key factor affecting microbiome profile. In this regard, i ask for major revision including more data as recommended from the first revision. 

AUTHORS’ REPLY.

Thank you to the Reviewer 3 for the further suggestions to revise and improve the manuscript.

  1. Concerning the average age differences between the ischemic stroke group (68.9 years) and healthy controls subjects (44.16 years) please consider our revised approach including new CTRL dataset age matched.

Based on your suggestions, we have expanded the CTRLs group database, please see the Methods section “2.4 16S rRNA Targeted Metataxonomy Analysis, by including other digital gut microbiota (GM) profiles belonging to older subjects available from our local BBMRI Human Microbiome Biobank, and characterized by our laboratory at Bambino Gesù Children’s Hospital in Rome.

Particularly, additional 15 new GM profiles were included to implement the group of healthy subjects (CTRLs), passing from 6 to 21 subjects. Based on this new extended CTRLs dataset we have obtained a final group of 21 aged-matched healthy subjects, but with a median age of 66.0 years ± 9.0 (IQR), more proximate to the average age of patients, as requested.

By considering the new CTRLs dataset the longitudinal metataxonomy (IS T0 and T1 vs CTRLs) and computational analyses were completely recomputed, and also the characterization of the IS profile at T0, at the onset of the disease, was compared to the new CTRLs dataset.

However, the metabolomic analyses were carried out only on the previous 6 selected CTRLs, as the new added control samples were not more available as biobank sample (i.e., insufficient quantity) after targeted-metagenomics processing, but only as digital GM profile.

2          Concerning the second point, regarding the possibility to include any additional data from other dataset worldwide, we did include this approach, also according to a previous publication from our group on ASD GM phenotype and batch effect (Pietrucci D, Teofani A, Milanesi M, Fosso B, Putignani L, Messina F, Pesole G, Desideri A, Chillemi G. Machine Learning Data Analysis Highlights the Role of Parasutterella and Alloprevotella in Autism Spectrum Disorders. Biomedicines. 2022 Aug 19;10(8):2028. doi: 10.3390/biomedicines10082028. PMID: 36009575; PMCID: PMC9405825).

Accordingly, to the request, we performed a deep literature searching with the aim to identify more studies related to IS disease, possibly including sequence databases to be considered as International repositories of sequencing data to be included in our analyses, as requested.

However, please consider that several criticisms, unfortunately, were encountered to address sequence data point.

In particular:

Several articles (n=7) NOT reported any information about sequence release into repository data (Sequences Read Archive database link and/or accession numbers). Hence, we tried to send an email to each corresponding Author, but unfortunately any feedback was obtained, as below documented (please see the email text sent to authors):

  • Xie H, Chen J, Chen Q, Zhao Y, Liu J, Sun J, Hu X. The Diagnostic Value of Gut Microbiota Analysis for Post-Stroke Sleep Disorders. Diagnostics (Basel). 2023 Sep 17;13(18):2970. doi: 10.3390/diagnostics13182970. PMID: 37761337; PMCID: PMC10530055.

No answer so far was received!

  • Li N, Wang X, Sun C, Wu X, Lu M, Si Y, Ye X, Wang T, Yu X, Zhao X, Wei N, Wang X. Change of intestinal microbiota in cerebral ischemic stroke patients. BMC Microbiol. 2019 Aug 19;19(1):191. doi: 10.1186/s12866-019-1552-1. PMID: 31426765; PMCID: PMC6700817

No answer so far was received!

  • Jiang, Z., M.M., Li, L., PhD., Liu, L., M.D., Ding, B., M.M., Yang, Y., M.M., He, F., M.M., Wu, Z., M.D. (2023). Ischemic stroke and dysbiosis of gut microbiota: Changes to LPS levels and effects on functional outcomes. Alternative Therapies in Health and Medicine, 29(5), 284-292. Retrieved from https://www.proquest.com/scholarly-journals/ischemic-stroke-dysbiosis-gut-microbiota-changes/docview/2838780070/se-2

No answer so far was received!

  • Tan C, Wu Q, Wang H, Gao X, Xu R, Cui Z, Zhu J, Zeng X, Zhou H, He Y, Yin J. Dysbiosis of Gut Microbiota and Short-Chain Fatty Acids in Acute Ischemic Stroke and the Subsequent Risk for Poor Functional Outcomes. JPEN J Parenter Enteral Nutr. 2021 Mar;45(3):518-529. doi: 10.1002/jpen.1861. Epub 2020 May 30. PMID: 32473086; PMCID: PMC8048557.

No answer so far was received!

  • Guo Q, Jiang X, Ni C, Li L, Chen L, Wang Y, Li M, Wang C, Gao L, Zhu H, Song J. Gut Microbiota-Related Effects of Tanhuo Decoction in Acute Ischemic Stroke. Oxid Med Cell Longev. 2021 May 27;2021:5596924. doi: 10.1155/2021/5596924. PMID: 34136066; PMCID: PMC8175183.

No answer so far was received!

  • Shi J, Zhao Y, Chen Q, Liao X, Chen J, Xie H, Liu J, Sun J, Chen S. Association Analysis of Gut Microbiota and Prognosis of Patients with Acute Ischemic Stroke in Basal Ganglia Region. Microorganisms. 2023 Oct 30;11(11):2667. doi: 10.3390/microorganisms11112667. PMID: 38004679; PMCID: PMC10673176.

No answer so far was received!

  • Li H, Zhang X, Pan D, Liu Y, Yan X, Tang Y, Tao M, Gong L, Zhang T, Woods CR, Du Y, Gao R, Qin H. Dysbiosis characteristics of gut microbiota in cerebral infarction patients. Transl Neurosci. 2020 Jun 8;11(1):124-133. doi: 10.1515/tnsci-2020-0117. PMID: 33312718; PMCID: PMC7706127.

No answer so far was received!

----------------------------------------------------------------------------------------------------------------------

For the following paper:

  • Liu Y, Kong C, Gong L, Zhang X, Zhu Y, Wang H, Qu X, Gao R, Yin F, Liu X, Qin H. The Association of Post-Stroke Cognitive Impairment and Gut Microbiota and its Corresponding Metabolites. J Alzheimers Dis. 2020;73(4):1455-1466. doi: 10.3233/JAD-191066. PMID: 31929168,

the pitfall regarded the fact that in the manuscript the links to supplementary materials were reported, only but available upon payment. For this reason, we sent an email to corresponding Author as follows:

But, no answer was so far received!

-----------------------------------------------------------------------------------------------------------------------

Moreover:

We found other articles in which the indication about sequence repository references data (Sequences Read Archive (SRA) database link and/or accession number) was present, but by searching with the accession number or link indicated in the articles the expected laboratory information for sequencing data generation were erroneously or even NOT reported or not matching with our experimental conditions. This was the case of:

  • Yamashiro K, Tanaka R, Urabe T, Ueno Y, Yamashiro Y, Nomoto K, Takahashi T, Tsuji H, Asahara T, Hattori N. Gut dysbiosis is associated with metabolism and systemic inflammation in patients with ischemic stroke. PLoS One. 2017 Feb 6;12(2):e0171521. doi: 10.1371/journal.pone.0171521. Erratum in: PLoS One. 2017 Apr 13;12 (4):e0176062. PMID: 28166278; PMCID: PMC5293236

Pitfall: The data uploaded on the repository SRA were obtained with targeted 16S and 23S rRNA qRT-PCR conducted using the Yakult Intestinal Flora-SCAN analysis system (YIF-SCAN®, Yakult Honsha Co., Ltd., Tokyo, Japan), therefore unusable for our bioinformatics analysis as the Authors used different pipeline compared to ours.

  • Xia GH, You C, Gao XX, Zeng XL, Zhu JJ, Xu KY, Tan CH, Xu RT, Wu QH, Zhou HW, He Y, Yin J. Stroke Dysbiosis Index (SDI) in Gut Microbiome Are Associated With Brain Injury and Prognosis of Stroke. Front Neurol. 2019 Apr 24;10:397. doi: 10.3389/fneur.2019.00397. PMID: 31068891; PMCID: PMC6491752.

Pitfall: The data uploaded on the repository SRA belonging to FASTA files and not FASTQ files, therefore unusable for our bioinformatics analysis

  • Chen J, Chi B, Ma J, Zhang J, Gu Q, Xie H, Kong Y, Yao S, Liu J, Sun J, Chen S. Gut microbiota signature as predictors of adverse outcomes after acute ischemic stroke in patients with hyperlipidemia. Front Cell Infect Microbiol. 2022 Nov 24;12:1073113. doi: 10.3389/fcimb.2022.1073113. PMID: 36506018; PMCID: PMC9729740.

Pitfall: The data uploaded on the repository SRA belonging to Long Reads PACBIO not derived from MiSeq Illumina technology as declared in the manuscript, hence unusable for our bioinformatics analysis.

  • Yin J, Liao SX, He Y, Wang S, Xia GH, Liu FT, Zhu JJ, You C, Chen Q, Zhou L, Pan SY, Zhou HW. Dysbiosis of Gut Microbiota With Reduced Trimethylamine-N-Oxide Level in Patients With Large-Artery Atherosclerotic Stroke or Transient Ischemic Attack. J Am Heart Assoc. 2015 Nov 23;4(11):e002699. doi: 10.1161/JAHA.115.002699. PMID: 26597155; PMCID: PMC4845212.

Pitfall: The data uploaded on the repository SRA do not contain the amplicon sequences hence unusable for our bioinformatics analysis.

  • Xu K, Gao X, Xia G, Chen M, Zeng N, Wang S, You C, Tian X, Di H, Tang W, Li P, Wang H, Zeng X, Tan C, Meng F, Li H, He Y, Zhou H, Yin J. Rapid gut dysbiosis induced by stroke exacerbates brain infarction in turn. Gut. 2021 Feb 8:gutjnl-2020-323263. doi: 10.1136/gutjnl-2020-323263. Epub ahead of print. PMID: 33558272.

Pitfall: The data were obtained only by amplification of the 16S rRNA gene V4 variable region but our dataset included the amplification of V3-V4 region, hence the retrieved data were unusable for our bioinformatics analysis.

However, we continued the searching of most updated articles describing IS GM profiling but only the following ones were retrieved as effective to consider an exploitable sequence dataset:

- Sun H, Gu M, Li Z, Chen X, Zhou J. Gut Microbiota Dysbiosis in Acute Ischemic Stroke Associated With 3-Month Unfavorable Outcome. Front Neurol. 2022 Jan 28;12:799222. doi: 10.3389/fneur.2021.799222. PMID: 35153980; PMCID: PMC8831883 referred to Chinese IS patients.

- Guo Q, Jiang X, Ni C, Li L, Chen L, Wang Y, Li M, Wang C, Gao L, Zhu H, Song J. Gut Microbiota-Related Effects of Tanhuo Decoction in Acute Ischemic Stroke. Oxid Med Cell Longev. 2021 May 27;2021:5596924. doi: 10.1155/2021/5596924. PMID: 34136066; PMCID: PMC8175183 referred to new Chinese CTRLs.

Hence, by exploiting sequence datasets from these papers we computed the batch effect, obtaining the following results:

For sure our GM metataxonomy profiles of IS patients were implemented with other data, as suggested.

Particularly, 52 GM profiles of fecal samples from Chinese patients with acute IS with good outcome, were added to our IS_T1 metataxonomy dataset profiles, similarly characterized by good clinical outcome, based on neurological features and clinical tests regarding the evaluation of functional movement, in order to compare more IS patients (n=62), arisen from different cohorts, with our new CTRLs (n=21) dataset, also implemented with new other 30 Chinese CTRLs (n=51).

Hence, the beta diversity was performed to evaluate the differences for the GM diversity between IS and CTRL groups.

Figure 1: Principal Coordinate Analysis (PCoA) plot shows Beta-diversity of either Italian and Chinese IS patients, and Italian and Chinese CTRLs performed by unweighted Bray- Curtis index.

The PERMANOVA test revealed a statistically significant difference (p-value=0.001) between IS patients and CTRLs. However, the PCoA showed a separation between Italian and Chinese IS patients, as highlighted in the Figure 1, where the blue dots on the right side represented only all Chinese IS patients. Moreover, the left side of Figure 1 showed a separation of CTRLs group between Italian (upper) and Chinese (bottom) cohort. This difference can be due, probably, to different variables (such as ethnicity, geographical origin, diet and lifestyle).

Despite the PCoA evidenced a partial overlap between Italian IS patients and Italian CTRLs (Figure 1), when we focusing only on the Italian population cohort, the GM diversity between CTRL group and IS patient groups, was clearly assessed (Figure 2). In particular, β-diversity computed by Bray-Curtis distance (Figure 2), showed the “intrinsic IS disease profile” patients at T0 separated from CTRLs (p-value=0.001, PERMANOVA). The same results were observed also in the comparison between IS at T1 and CTRLs, as shown in the main text R2 version (Figure 1 in the Results Section), with no significant differences in GM ecology between IS at T0 and IS at T1.

Figure 2: GM ecological analysis: Principal Coordinate Analysis (PCoA) plots shows Beta-diversity of IS T0 and CTRLs performed by Bray- Curtis index.

IS_T0: IS patients at T0; CTRL: controls subjects.

In particular, the LEfSe analyses, performed on the ASVs of IS patients at T0 and CTRLs corroborated the longitudinal analyses, described in the “3.2. Gut microbiota metaxonomy” paragraph.  The assay confirmed the ASVs associated to the IS disease through the longitudinal time points (T0 and T1), namely Bacteroides, Akkermansia, Alistipes, Christensellaceae (Figure 3 A).

In addition, in order to highlight the functional differences between IS at T0 and CTRLs, the KEGG metabolic pathways were predicted by performing by PICRUSt2 (Figure 3 B).

Figure 3: (A) LEfSe analysis for the comparison between the IS patients at T0 and CTRLs groups’ GM. Histogram represents the LDA scores until L6 taxonomy filtered by statistical significance between the two groups. In red and in green are represented microbial biomarkers for the CTRLs subjects and IS patients, respectively; (B) Kyoto Encyclopedia of Genes and Genomes (KEGG) biomarkers inferred from the whole set of 46 ASVs of the IS patients at T0 and the CTRLs. Linear discriminant analysis (LDA) effect size (LEfSe) was performed on the PICRUSt2 predicted biochemical pathways matrix. The reported pathways were filtered for statistically significance and LDA ± 3.0.

IS_ T0: IS patients at T0; CTRL: controls subjects; LDA: Linear Discriminant Analysis score.

LEfSe analyses performed also on ASVs and KEGG to compare IS patient at T1 and CTRLs showed the same significative results for IS at T0.

Therefore, these results highlighted both ecological and functional differences between Italian IS patients and Italian CTRLs. 

In conclusion, our results show a specific IS disease-related profile, although specific differences among GM profiles depending from geography, ethnicity, and probably different diet behavior were observed. For this reason, combined to the modifications of the text in this R2 revision, we propose a new title: “Gut microbiota ecological and functional modulation in post-stroke recovery patients: an Italian study”, to take into account the possible variability of IS profiles worldwide.
